# The Role of Flagellum and Flagellum-Based Motility on *Salmonella* Enteritidis and *Escherichia coli* Biofilm Formation

**DOI:** 10.3390/microorganisms12020232

**Published:** 2024-01-23

**Authors:** Diana Vilas Boas, Joana Castro, Daniela Araújo, Franklin L. Nóbrega, Charles W. Keevil, Nuno F. Azevedo, Maria João Vieira, Carina Almeida

**Affiliations:** 1Center of Biological Engineering (CEB), Campus de Gualtar, University of Minho, 4710-057 Braga, Portugal; dianavilasboas@ceb.uminho.pt (D.V.B.); mjv@deb.uminho.pt (M.J.V.); 2LABBELS–Associate Laboratory, Braga/Guimarães, 4710-057 Braga, Portugal; 3INIAV—National Institute for Agrarian and Veterinarian Research, Rua dos Lagidos, 4485-655 Vila do Conde, Portugal; joana.castro@iniav.pt (J.C.); daniela.araujo@iniav.pt (D.A.); 4School of Biological Sciences, University of Southampton, University Road Southampton, Southampton SO17 1BJ, UK; f.nobrega@soton.ac.uk (F.L.N.); c.w.keevil@soton.ac.uk (C.W.K.); 5LEPABE—Laboratory for Process Engineering, Environment, Biotechnology and Energy, Faculty of Engineering, University of Porto, Rua Dr. Roberto Frias, 4200-465 Porto, Portugal; nazevedo@fe.up.pt; 6AliCE—Associate Laboratory in Chemical Engineering, Faculty of Engineering, University of Porto, Rua Dr. Roberto Frias, 4200-465 Porto, Portugal

**Keywords:** biofilm formation, foodborne pathogens, *Salmonella*, *Escherichia coli*, fliC mutants, *motA* mutants

## Abstract

Flagellum-mediated motility has been suggested to contribute to virulence by allowing bacteria to colonize and spread to new surfaces. In *Salmonella enterica* and *Escherichia coli* species, mutants affected by their flagellar motility have shown a reduced ability to form biofilms. While it is known that some species might act as co-aggregation factors for bacterial adhesion, studies of food-related biofilms have been limited to single-species biofilms and short biofilm formation periods. To assess the contribution of flagella and flagellum-based motility to adhesion and biofilm formation, two *Salmonella* and *E. coli* mutants with different flagellar phenotypes were produced: the *fliC* mutants, which do not produce flagella, and the *motAB* mutants, which are non-motile. The ability of wild-type and mutant strains to form biofilms was compared, and their relative fitness was determined in two-species biofilms with other foodborne pathogens. Our results showed a defective and significant behavior of *E. coli* in initial surface colonization *(p* < 0.05*)*, which delayed single-species biofilm formation. *Salmonella* mutants were not affected by the ability to form biofilm (*p* > 0.05). Regarding the effect of motility/flagellum absence on bacterial fitness, none of the mutant strains seems to have their relative fitness affected in the presence of a competing species. Although the absence of motility may eventually delay initial colonization, this study suggests that motility is not essential for biofilm formation and does not have a strong impact on bacteria’s fitness when a competing species is present.

## 1. Introduction

Motile bacteria, such as *Escherichia coli* and *Salmonella*, have the ability to move through liquid medium by rotating peritrichous helical flagella that extend from their cell surface [1]. A bacterial flagellum is a macromolecular machine that consists of a basal body (rotary motor), hook (universal joint), and filament (propeller) [2]. Flagellar biosynthesis is a process that requires 2% of the cell’s biosynthetic resources [3]. Genes for flagellum synthesis are usually organized into a cluster that encodes transcription factors, basal body and hook proteins of the flagellum, the filament of the flagellum (FliC), as well as proteins necessary for motility and chemotaxis [4].

Flagellum is known to be a key factor in facilitating the initial contact of the cell with the surface and early biofilm formation in several Gram-negative bacteria [5,6,7,8]. It has been proposed that flagellar activity may help to overcome repulsive forces at the surface, thereby allowing initial surface contact [5]. In several bacteria, including *E. coli*, the flagellum has a multifunctional role, participating in motility and acting as an adhesive organelle [6].

To better understand flagellum-based motility and biofilm formation, some authors have produced different mutants that affect motility. In *E. coli*, cells that either lack complete flagella or have paralyzed flagella are severely hindered in the early stages of biofilm formation, indicating that motility is the key factor in early biofilm formation [8,9,10]. Also, the biofilm formed consists of isolated microcolonies (small, dense clusters of cells instead of homogeneous distribution), which suggests that once the *E. coli* cells are close to the surface, flagellum-mediated motility is required for movement parallel to the surface (in addition to bringing the bacteria into proximity to the surface) [5]. In *Salmonella* Typhimurium, previous studies with *motA* and *fliA* mutants have shown that motility is a prerequisite for biofilm formation on glass surfaces, and *Salmonella* mutants with impaired flagella are unable to form fully developed biofilms [11,12]. This suggests that the role of the flagellum may depend not only on the species but also on the physical chemistry of the surface used for cell adhesion. However, studies of the role of the flagellum have been limited to one or two materials, and nothing is known about the effect of motility and flagellum on mature stages of single or mixed biofilm formation. This may not provide a correct picture of the role of the flagellum in complex communities. In fact, it is known that most natural biofilms consist of complex communities and that these structures are typically characterized by the presence of primary and later colonizers [13,14]. Neighboring species may help with initial colonization, as is the case with well-studied oral biofilms, or they may eventually compete for surface space [13]. Whether flagellum/motility is required for adhesion and biofilm formation in mixed communities or whether flagellum-affected species have reduced overall fitness is still an understudied subject.

To gain a broader perspective on the link between flagellum/flagellum-based motility and biofilm development, *Salmonella* Enteritidis and *E. coli* mutants with paralyzed flagellum (Δ*motA*) and lacking flagellum (Δ*fliC*) were produced and evaluated for their initial adhesion and biofilm formation on different surfaces. Then, their fitness was determined by the presence and absence of selected competing species.

## 2. Materials and Methods

### 2.1. Bacterial Strains and Growth Conditions

*E. coli* CECT 434, *Salmonela enterica* serovar Enteritidis ATCC 13076, *Listeria monocytogenes* CECT 4031T, and *Staphylococcus aureus* CECT 239 were maintained on Tryptic Soy Agar (TSA) (VWR, Carnaxide, Portugal), grown overnight at 37 °C, and streaked onto fresh plates every 24 h. It is worth noting that *L. monocytogenes* and *S. aureus* were used in fitness dual-species biofilms as competing species. For the preparation of the inoculum, cells were grown overnight (approximately 16 h) in Tryptic Soy Broth (TSB) (VWR) at 37 °C and 120 rpm. Throughout the manuscript, to simplify the description of the results, “WT” refers to wild-type, “Δ*flic*” to the mutant without flagellum, and “Δ*motA*” to the mutant with a paralyzed flagellum.

### 2.2. Plasmids

Template plasmid pKD4, carrying a *KnR* gene flanked by FRT (FLP recognition target) sites, was used to generate the polymerase chain reaction (PCR) fragments used for homologous recombination. The plasmid was kindly provided by Dr. Sara Cleto (formerly associated with Harvard Medical School).

pKD46, red helper plasmid, is an ampicillin and CmR plasmid that shows temperature-sensitive replication [15]. This plasmid includes *γ*, *β*, and *exo* genes (λ red recombinase) under the control of an arabinose-inducible P_araB_ promoter. Gam inhibits the host RecBCD exonuclease V so that Bet and Exo can gain access to DNA ends to promote recombination.

The FLP helper plasmid pCP20, encoding the FLP recombinase, was used to eliminate the resistance genes after homologous recombination. FLP recombinase acts on the directly repeated FRT (FLP recognition target) sites flanking the resistance gene. The red and helper plasmids are temperature-sensitive replicons and were cured by growth at 37 °C.

### 2.3. Gene Disruption

Lamba red-mediated recombination, as described by Datsenko and Wanner [16], was used. The primers were designed using the Primer 3 web-based software (http://www.bioinformatics.nl/cgi-bin/primer3plus/primer3plus.cgi accessed on January 2021) and are described in Table 1. As such, primers used for mutant construction and confirmation are described in Table 1. It is important to note that mutations were confirmed by PCR.

*E. coli* and *S.* Enteritidis were first transformed with the pKD46 plasmid. Cells were grown at 37 °C to an optical density (OD)_600nm_ = 0.6 and then made electrocompetent by concentrating 100-fold and washing three times with ice-cold 10% (*v*/*v*) of glycerol. Electroporation was performed using an Electroporator Pulser^TM^ Transformation Apparatus (Bio-Rad Laboratories, Hercules, CA, USA) with a voltage booster and 0.15 cm chambers according to the manufacturer’s instructions. Fifty µL of cells and 10–100 ng of pKD46 plasmid, were used. One mL of super-optimal broth culture medium (SOC, Thermo Fisher Scientific, Waltham, MA, USA) was added, and cells were incubated for 1 h at 37 °C and then spread onto agar to select ampicillin-resistant (AmpR) transformants. It is worth noting that AmpR transformants were selected on tryptone-yeast extract agar medium (TY, Liofilchem, Roseto degli Abruzzi, Italy) containing ampicillin at 100 mg/mL. Next, the *E. coli* and *S.* Enteritidis strains carrying a red helper plasmid pKD46 were grown in 5 mL super-optimal broth medium (SOB, Thermo Fisher Scientific) cultures with ampicillin and L-arabinose at 30 °C to reach an OD_600nm_ of 0.6 and then made electrocompetent as described above. PCR products were purified, digested with DpnI (New England Biolabs, Ipswich, MA, USA), repurified, and suspended in an elution buffer (10 mM Tris, pH 8.0). Electroporation was performed using 50 µL of cells and 10–100 ng of PCR product. Shocked cells were added to 1 mL of SOC and incubated for 1 h at 37 °C, and then one half was spread onto an agar to select KmR transformants. As such, KmR-resistant transformants were selected on TY agar containing kanamycin at 50 mg/mL. If non-growth was observed within 24 h, the remainder was spread after incubation overnight at room temperature. After primary selection, mutants were maintained on a medium without antibiotics. They were non-selectively colony-purified once at 37 °C and then tested for ampicillin sensitivity to test for loss of the helper plasmid. If the plasmid was not lost, a few colonies were colony-purified at 43 °C and tested again.

### 2.4. Eliminating Antibiotic Resistance Gene

The FLP recombinase encoded by the pCP20 plasmid was used to eliminate the antibiotic resistance cassette. More precisely, KmR mutants were transformed with pCP20, and ampicillin-resistant transformants were selected at 30 °C, after which a few were non-selectively colony-purified at 43 °C and then tested for loss of all antibiotic resistances [16]. The pCP20 plasmid encodes the Flp recombinase that catalyzes the recombination between the FRT sites flanking the antibiotic resistance cassettes. In this way, it is possible to ensure that resistance is eliminated from the strain, ensuring a “clean” knockout. As such, the strains simultaneously lost the FRT-flanked resistance gene and the FLP helper plasmid. Finally, cells were grown at TY to induce FLP action and remove the resistance. The gene interruptions were confirmed by PCR using the primers listed in Table 1.

### 2.5. The Effect of motA or fliC Deletion on S. Enteritidis and E. coli Swarming

The effect of *motA* or *fliC* deletion on *S.* Enteritidis and *E. coli* motility was assessed through the evaluation of swarming, as described in a previous study [17]. To test motility, a sterile needle was used to lightly touch an overnight *S.* Enteritidis and *E. coli* culture and spotted gently in the middle of a swarm plate (Nutrient Broth (NB, Liofilchem), 0.5% (*w*/*v*) glucose (Liofilchem), 0.6% (*w*/*v*) bacteriological agar (Liofilchem)). The plates were incubated at 37 °C for 24 h. The results are the means of at least three independent experiments.

### 2.6. Biofilm Formation

The cultures were first grown overnight (16 to 18 h) in TSB (VWR) at 37 °C and 120 rpm. After that, the inoculum concentration was adjusted to 1 × 10^6^ CFU/mL. After homogenization, 6 mL of the suspension was dispensed into each well of a six-well tissue culture plate (Orange Scientific, Braine L’Alleud, Belgium) containing coupons (dimensions of 2 × 2 cm) of different materials (glass, polypropylene (PP), and stainless steel (steel)), prepared as previously described [18]. To ensure sterility, coupons were autoclaved for 20 min at 120 °C. The tissue culture plates were then placed in an incubator (Shell Lab, Hillsboro, OR, USA) at 21 °C in standing culture. At different sampling times (2, 4, 6, 24, and 48 h), coupons were removed from the tissue plates, washed twice with 6 mL of fresh Phosphate Buffered Saline (PBS) to remove the loosely attached cells, and biofilm formation was assessed by plate counts of colony-forming units (CFUs), crystal violet (CV) assay, and DAPI (4,6-diamidino-2-phenylindole) staining as described below. For dual-species biofilms, 3 mL of the bacteria suspensions at a concentration of 2 × 10^6^ CFU/mL was mixed and placed on the six-well plate in order to obtain a final concentration of 10^6^ CFU/mL. Then, experiments followed the same workflow described above. Biofilm assays were repeated independently at least three times with technical replicates.

### 2.7. Cultivability Assessment

To determine the amount of bacteria present in biofilms, CFU counts were performed using the microdrop technique. Briefly, after washing, the coupons with biofilm were placed in a new six-well tissue culture plate with 6 mL of PBS and sonicated with a 5-s burst at 25% amplitude using a GEX 400 ultrasonic processor (Sigma-Aldrich, St. Louis, MO, USA). Next, 100 μL samples were taken to assess cultivability by plating the appropriate dilutions on agar plates in triplicate. Plates were incubated at 37 °C for 16–18 h (*E. coli* and *S.* Enteritidis), 24 h (*S. aureus*), and 48 h (*L. monocytogenes*); then, CFUs were counted. For better discrimination, the following three selective agar media were used: MacConkey agar (Liofilchem) for distinguished *E. coli* and *S.* Enteritidis, Mannitol salt agar (Liofilchem) for *S. aureus,* and Listeria oxford agar (Liofilchem) for *L. monocytogenes.* The number of cultivable bacterial cells in biofilms was determined and expressed per area of coupons (log CFU/cm^2^). To evaluate the recovery ability of the selective medium, a control test was performed by plating known concentrations on TSA and the corresponding selective media. No significant differences were found between the CFU counts in TSA and the selective/differential medium used.

### 2.8. Biomass Quantification by the CV Assay

The quantification of biofilm biomass production was based on the previously described method [19]. The washed coupons were placed in a new six-well tissue culture plate and fixed with 3 mL of methanol 98% (*v*/*v*) (Thermo Fisher Scientific) for 15 min. Afterward, the methanol was removed, and the coupons were allowed to air-dry. Then, biofilms were stained with 3 mL of CV (Merck, Darmstadt, Germany) for 5 min. Coupons were washed three times by pouring tap water over them, allowed to air-dry, and then the CV was removed by adding 6 mL of 33% (*v*/*v*) glacial acetic acid (Merck) to each well. The plates were placed in agitation for a few minutes, and 250 μL were transferred to a 96-well microtiter plate. Subsequently, the OD was measured at 570 nm using a microtiter plate reader (Model Sunrise, Tecan, Tokyo, Japan). Biofilm assays were repeated three times on separate days.

### 2.9. Determination of the Fitness and Malthusian Parameter

The fitness determined for each dual-species biofilm was estimated as the ratio of the Malthusian parameters of each population. The Malthusian fitness parameter (*m_i_)* can be determined as the average rate increase was calculated over time: m = ln [N_i_(T_final_)/N_i_(T_inicial_)]/t_final_, where N_i_ is the population density value (CFU/cm^2^) present in the biofilm at the initial and final time points [20]. The time measured is specified by t. The relative fitness was determined here as the selection rate constant, r_ij_ = *m_i_* − *m_j_*, resulting in a fitness of zero when competing species are equally fit.

### 2.10. PNA FISH Hybridization and DAPI Staining

Biofilm cells were discriminated against using specific and previously developed PNA probes for the detection of *S.* Enteritidis (SalPNA1873) [21] and *E. coli* [22]. Briefly, 6 h biofilm coupons were taken, washed in PBS, and covered with methanol for 10 min. Then, 4% (*w*/*v*) paraformaldehyde (Thermo Fisher Scientific) and 50% (*v*/*v*) of ethanol (Thermo Fisher Scientific) were added separately for 10 min each. Subsequently, 100 μL of hybridization solution containing 10% (*w*/*v*) dextran sulphate (Sigma-Aldrich), 10 mM NaCl, 30% (*v*/*v*) formamide (Sigma-Aldrich), 0.1% (*w*/*v*) sodium pyrophosphate (Sigma-Aldrich), 0.2% (*w*/*v*) polyvinylpyrrolidone (Sigma-Aldrich), 0.2% (*w*/*v*) Ficol, 5 mM disodium EDTA (Sigma-Aldrich), 0.1% (*v*/*v*) Triton X-100, 50 mM Tris-HCl (pH 7.5) with 200 nM of PNA probe. The samples were incubated at 57 °C for 30 min. After hybridization, coupons were placed into a wash solution containing 5 mM Tris Base (Thermo Fisher Scientific), 15 mM NaCl (Liofilchem), and 1% (*v*/*v*) Triton X (Fisher Bioreagents, Pittsburgh, PA, USA, pH 10), and incubated at 57 °C for 30 min. Samples were allowed to air-dry, mounted with one drop of nonfluorescent immersion oil (Merck), and covered with coverslips. Cells were visualized under an epifluorescence microscope (BX51; Olympus, Tokyo, Japan) equipped with a CCD camera (DP71; Olympus) and filters capable of detecting the two PNA probes (BP 530–550, FT 570, LP 591 sensitive to the Alexa Fluor 594 molecule attached to the SalPNA1873 probe; BP 470–490, FT 500, LP 516 sensitive to the Alexa Fluor 488 molecule attached to the Ec1505 LNA/2 OMe 16 [22]) and DAPI (BP 365–370, FT 400, LP 421).

### 2.11. Statistical Analysis

Results were compared using One-Way analysis of variance (ANOVA) by applying Levene’s test of homogeneity of variance and the Tukey multiple-comparisons test using Microsoft Office Excel 2021 (Microsoft Corporation, Redmond, CA, USA). All tests were performed with a confidence level of 95%. Values with a *p* < 0.05 were considered statistically significant.

## 3. Results

### 3.1. Flagellum and Flagellum-Based Motility Effect on S. Enteritidis and E. coli Biofilm Formation

The presence of flagellum or flagellum-based motility has long been associated with the strains’ ability to colonize different surfaces, but the effect of the surface properties on this ability remains elusive so far. To better clarify this issue, mutant *S.* Enteritidis and *E. coli* strains lacking either flagellum or motility were produced to further evaluate their ability to adhere to surfaces with different properties. Genes coding for Flagelin (*fliC*), the flagellum filament, and a motor protein (*motA*), were targeted using the Datsenko and Wanner method [16]. The mutant genotype was confirmed by PCR, and the phenotype was confirmed in motility agar (Figure 1). In fact, according to Figure 1, no swarming was observed for *S.* Enteritidis and *E. coli* mutant strains when compared with WT strains.

After confirming the absence of motility, mutant and WT strains were allowed to form biofilm on three different surfaces: glass, steel, and PP (please see Appendix A for further details on surface properties).

Regarding their ability to adhere to different surfaces, no significant differences were found between the ∆*fliC* and ∆*motA* mutants for both species, which seems to indicate that motility, and not flagella, is likely the key factor; otherwise, a difference would be noticed between the two mutants. Similarly, no differences were found between the surfaces analyzed (see Figure 2).

Concerning the behavior of mutant versus WT strains, for *S.* Enteritidis*,* no significant differences were found in the initial adhesion according to the cultivability (Figure 3A) and biofilm biomass (Figure 3C) findings, while for *E. coli,* the initial adhesion of mutant strains seems to be affected, with a one to three log reduction being observed for the first time points (2, 4, and 6 h) (Figure 3B). Here, for *E. coli*, the absence of motility or flagellum only seems to cause a delay in cell adhesion, but after overcoming this delay, the biofilm population seems to reach the same levels as those observed for the WT strain (Figure 3B). Nonetheless, this delay in biofilm formation seems to also be reflected in the total biomass that remains low at 48 h of biofilm formation for the mutant strains (Figure 3D).

This behavior was also confirmed under the microscope by analyzing biofilms of the mutant and WT strains after 6 h (Figure 4).

Overall, while for *E. coli,* motility was particularly important for initial adhesion, for later stages of biofilm formation, the lack of motility did not represent an important disadvantage for *E. coli* in terms of cultivability. In the absence of flagellum or flagellum-mediated motility, *E. coli* is able to establish stable attachment to abiotic surfaces. Nonetheless, flagellin did not seem to be involved as an adhesin.

### 3.2. Salmonella and E. coli Mutants Fitness in Dual-Species Biofilms

The results obtained in the present work for single-species biofilms have suggested that the flagellum or flagellum-mediated motility is not mandatory for biofilm formation. However, the fact is that this might be a valuable element when considering complex communities with special limitations that impose competition for space. In contrast, it is also known that some species can take advantage of primary colonizers by using them as bridges for attachment [23]. To better understand if motility and flagellum can somehow influence the bacteria’s fitness in complex communities, dual-species biofilms were formed and evaluated over 96 h, so the strains’ fitness could be estimated from early to mature biofilm stages. The relative fitness of *E. coli* and *S.* Enteritidis (WT and each mutant strain) was determined in combination with other common food-related species, namely, *S. aureus*, *L. monocytogenes*, and *Salmonella* (WT strain) and/or *E. coli* (WT strain). In the presence of *L. monocytogenes*, the fitness of *E. coli* WT and mutant strains decreased during the first 48 h and remained zero after this time, meaning that both bacteria are equally fit (Figure 5A). In contrast, in the presence of *S. aureus* and *S.* Enteritidis, the fitness of *E. coli* increased during the first 24 h, after which the fitness of both species remained equal (Figure 5A). This is the case except for combination with the ∆*fliC* mutant and *S.* Enteritidis, where the fitness of *E. coli* decreased over 24 h. In the case of *S.* Enteritidis in the combination of other species, the fitness profile was similar for all three species tested, decreasing over the first 48 h, and then the bacteria were equally fit (Figure 5B). The only exception is the combination of the ∆*fliC* mutant and *E.coli* WT, where fitness increased during the first 48 h. Overall, in the presence of *L. monocytogenes*, the competition with *E. coli* and *S.* Enteritidis decreases. In contrast, when both species compete with *S. aureus*, the fitness profile is reversed, with competition with *E. coli* increasing against *S. aureus.* In general, no differences were observed when comparing WT strains and mutant strains (∆*fliC* and ∆*motA*).

## 4. Discussion

Extensive research over decades has focused on the motility of bacterial flagella, leading to significant advances in the understanding of the underlying mechanisms [5]. In fact, flagellum-driven motility has been proposed as a factor contributing to the virulence of bacteria by facilitating their capacity to inhabit and spread to new surfaces [10]. It has been described that bacterial mutations affecting flagellar motility have resulted in reduced abilities to create biofilms. Although certain species are recognized to act as co-aggregation factors promoting bacterial adhesion, investigations into food-related biofilms have been restricted to single-species biofilms and brief periods of biofilm formation [24,25]. Here, to assess the role of the flagellum and flagellum-based motility in adhesion and biofilm formation, two types of mutants with distinct flagellar phenotypes were generated for *S.* Enteritidis and *E. coli: fliC* mutants, which do not generate flagella, and non-motile *motAB* mutants. The biofilm-forming capabilities of both WT and mutant strains were compared, and their relative fitness was gauged in two-species biofilms alongside other foodborne pathogens.

It is noteworthy that diverse approaches have been devised to prevent biofilm formation in the early stages [26]. Studies on bacterial adhesion have revealed that, apart from the physicochemical surface characteristics of both the bacterium and substratum, biological elements contribute to the initial stages of adhesion. This implies that depending solely on thermodynamic methods may not precisely forecast adhesive capabilities. Most surfaces employed in the food industry consist of materials such as stainless steel, polyethylene, polypropylene, polycarbonate, carbon steel, fiberglass, polyurethane, polyvinyl chloride (PVC), marble, silicone, granite, Teflon, or glass [27]. As such, herein, ∆*fliC* and ∆*motA* mutants and WT strains were allowed to form biofilm on three different common surfaces used in the food industry field, namely: steel, glass, and PP. Regarding their ability to adhere to these different surfaces, no significant differences were found between the surfaces analyzed (Figure 2).

Concerning the behavior of mutant versus WT strains, our results demonstrated impaired initial surface colonization behavior in *E. coli*, leading to a delayed formation of single-species biofilms. These findings are supported by Benyoussef and colleagues, who have shown that motility introduces a significant delay in the colonization of bare surfaces by *E. coli* alone [5]. Furthermore, Huber and coworkers showed that biofilms produced by *E. coli motA* mutants exhibit significantly reduced surface-associated biomass compared to WT biofilms in the first 8 h, but within 48 h, the biofilms of the WT strain and the *motA* mutant strain are virtually indistinguishable [28]. In our study, we also verified the same tendency in biofilm formation when we observed the cultivable cells from the biofilm (Figure 3B). However, when we observed the biofilm biomass, we observed a lower biofilm biomass of mutant *E. coli* strains when compared with WT. This result suggests that the mutant strains might affect the production of biofilm matrix.

Contrary to what was reported in the literature [29], in our study, *S.* Enteritidis ∆*fliC* and ∆*motA* mutants did not exhibit any impact on their ability to form biofilm, even in the first stages. Wang and colleagues [29] in a study that constructed flagella mutants (∆*flgE* and ∆*fliC*) for *Salmonella enterica* Serovar Typhimurium reported that these mutants lacking flagellar motility form fewer biofilms in the early stage, and the mature biofilms that form contain more cells and extracellular polymeric substances (EPS). In addition, Prouty and Gunn [30] also examined the other mutants related to the *fliA* and *motA* mutants of *Salmonella enterica* Serovar Typhimurium biofilm formation on glass coverslips. According to the authors, these mutants caused a deficiency in biofilm formation and concluded that motility is required for biofilm development on glass. However, using our experimental conditions, such differences were not found.

In general, the interaction between *E. coli* and other species (*S. aureus* and *S.* Enteritidis) increased over time, remaining equally fit after 24 h. In another study, the relative fitness of *E. coli* was determined in dual-species biofilms, and it was observed that there was a slight increase in the fitness of *E. coli* over time [20]. In contrast, the fitness of *S.* Enteritidis decreases in the presence of other species, remaining equally fit after 48 h. Moreover, in terms of the effect of motility or flagellum absence on bacterial fitness, none of the mutant strains appeared to experience a significant decline in relative fitness when in the presence of a competing species. Although the absence of motility could potentially postpone initial colonization, this study suggests that motility is not indispensable for biofilm formation and has a limited impact on bacterial fitness in the presence of a competing species.

## 5. Conclusions

The role that flagellum-mediated motility plays in biofilm formation differs between species, even for related bacteria such as *E. coli* and *S.* Enteritidis. First, no significant differences were found between the ∆*fliC* and ∆*motA* mutants for both species; each might indicate a more important role of motility for itself than flagellum-based adhesion. Similarly, no differences were found between the surfaces analyzed independently of the hydrophobic/hydrophilic properties. As a general behavior, the lack of motility did not hinder biofilm formation for both *E. coli* and *S.* Enteritidis, but it clearly limited *E. coli* adhesion to abiotic surfaces (a behavior not observed for *S.* Enteritidis). The absence of motility caused a delay in initial colonization, especially for *E. coli*. On the other hand, flagellin did not seem to be involved as an adhesin in biofilm formation on abiotic surfaces. Concerning the impact of the absence of motility or flagellum on bacterial fitness, none of the mutant strains appear to experience a decline in their relative fitness in the presence of a competing species. Although the lack of motility may temporarily postpone the initial colonization, this investigation seems to indicate that motility is not a prerequisite for biofilm formation and might not exert a significant influence on bacterial fitness when there is competition from another species.

## Figures and Tables

**Figure 1 microorganisms-12-00232-f001:**
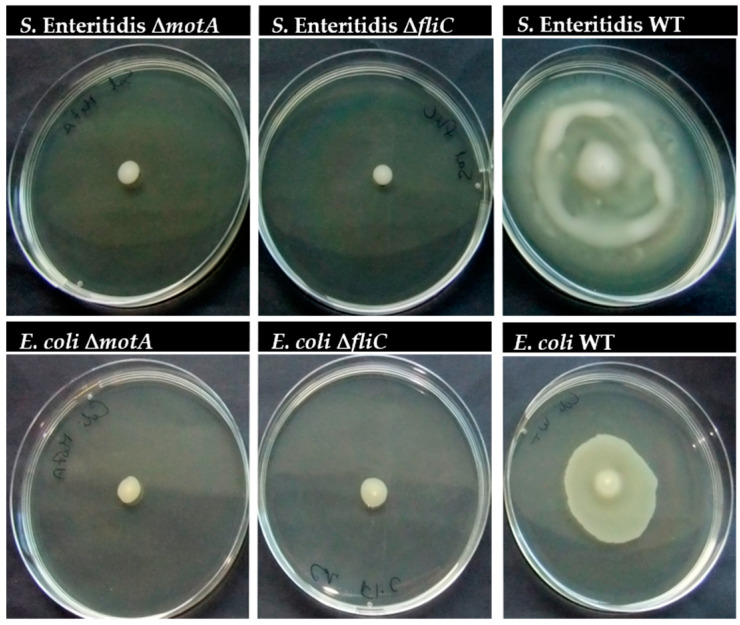
Effect of *motA* or *fliC* deletion on *S.* Enteritidis and *E. coli* motility. Bacteria were incubated overnight on a motility agar medium (0.6% (*w*/*v*) agar) to evaluate swarming.

**Figure 2 microorganisms-12-00232-f002:**
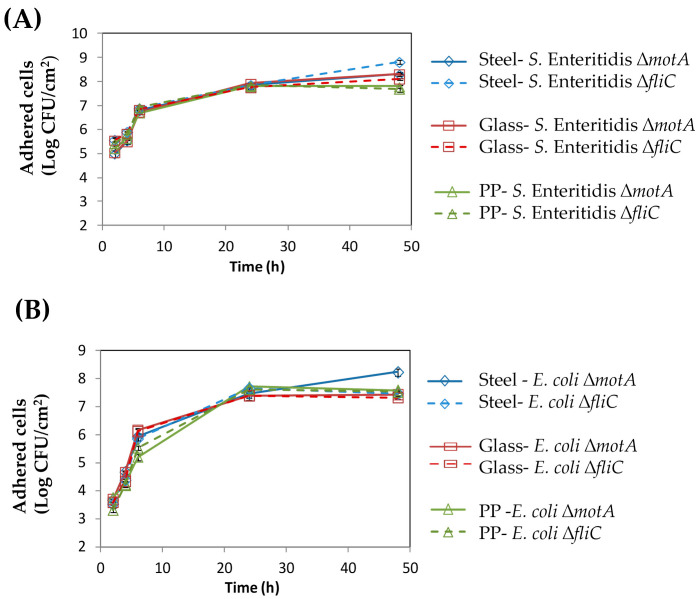
Biofilm formation of the *S.* Enteritidis (**A**) and *E. coli* mutants (**B**) on three different surfaces (steel, glass, and polypropylene (PP)).

**Figure 3 microorganisms-12-00232-f003:**
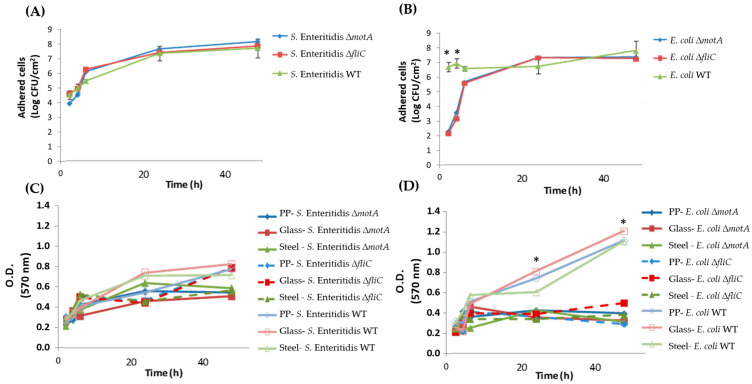
Biofilm formation of *S.* Enteritidis and *E. coli* WT and mutant strains. Cultivability only on glass surfaces for (**A**) *S.* Enteritidis and (**B**) *E. coli.* Biofilm biomass on three different surfaces for (**C**) *S.* Enteritidis and (**D**) *E. coli.* Abbreviations*:* WT: wild-type. * *p* < 0.05 as determined by the Tukey test.

**Figure 4 microorganisms-12-00232-f004:**
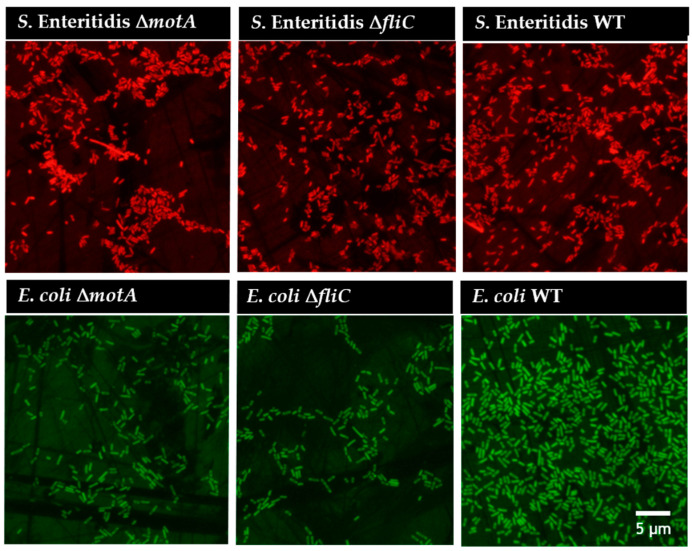
Epifluorescent images of *S.* Enteritidis and *E. coli* mutant and WT strains for a 6 h biofilm. The first line shows *S.* Enteritidis and the second line shows *E. coli*. The first and second columns show the mutant strains, and in the last column, WT strains. All images have the same scale (5 µm) and a magnification of 1000×.

**Figure 5 microorganisms-12-00232-f005:**
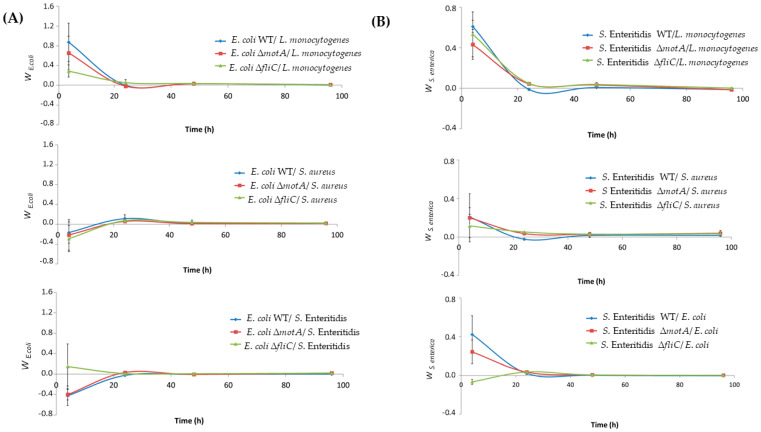
Relative fitness of *E. coli* and *S.* Enteritidis in dual-species biofilms. (**A**) Fitness of *E. coli* was determined in the presence of *L. monocytogenes*, *S. aureus*, and *S.* Enteritidis. (**B**) Fitness of *S.* Enteritidis was determined in the presence of *L. monocytogenes*, *S. aureus*, and *E. coli*. Data are means of three independent experiments, and error bars represent the SD.

**Table 1 microorganisms-12-00232-t001:** Primers used in this study to generate and confirm the mutants.

	Primer	Sequence (5′-3′) *
		Recombination primers
*E. coli*	FliC_Fw	GCAGCAGAGACAGAACCTGCTGCGGTACCTGGTTAGCTTTGTGTAGGCTGGAGCTGCTTC
	FliC_Rv	ATGGCACAAGTCATTAATACCAACAGCCTCTCGCTGATCAATGGGAATTAGCCATGGTCC
	MotAB_Fw	GTTTTCATGCAAAATGGCCTGTTCGGCTTGTTTGTTCAGTGTGTAGGCTGGAGCTGCTTC
	MotAB_Rv	GTGCTTATCTTATTAGGTTACCTGGTTGTTCTCGGTACAGATGGGAATTAGCCATGGTCC
*S.* Enteritidis	FljB_Fw	TTAACGCAGTAAAGAGAGGACGTTTTGCGGAACCTGGTTGTGTAGGCTGGAGCTGCTTC
	FljB_Rv	ATGGCACAAGTCATTAATACAAACAGCCTGTCGCTGTTGAATGGGAATTAGCCATGGTCC
	MotBA_Fw	GTTTTCATGCAAAATGGCCTGTTCCGCCTGTTTGTTTAACGTGTAGGCTGGAGCTGCTTC
	MotBA_Rv	GTGCTTATCTTATTAGGTTACCTGGTGGTTATCGGTACAGATGGGAATTAGCCATGGTCC
		Confirmation primers
*E. coli*	FliC Fw	TCAGGCAATTTGGCGTTGCCGTC
	FliC Rv	CAGACGATAACAGGGTTGACGGC
	MotAB Fw	GCCAACAGTTCGTCCGCTTC
	MotAB Rv	CTGTCATGGTCAACAGTGGAAG
*S.* Enteritidis	FliC Fw	GCAGGTTCAGTGACGGTGATT
	FliC Rv	CGAAATTCAGGTGCCGATACA
	MotBA Fw	GCACCAAATCCAGCAGATGCTG
	MotBA_Rv	TTGCCTTGCCTTCGCGTTAATC

Note. * underlined nucleotides on recombination primers correspond to priming sequences.

## Data Availability

Data are contained within the article.

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
