# Peer review of "The Role of Flagellum and Flagellum-Based Motility on Salmonella Enteritidis and Escherichia coli Biofilm Formation"

_microorganisms, 2024, doi:10.3390/microorganisms12020232_

Round 1

Reviewer 1 Report

Comments and Suggestions for Authors

I hope this message finds you well. I am writing regarding the recent review of the manuscript titled “The role of flagellum and flagellum-based motility on  Salmonella Enteritidis and Escherichia coli biofilm formation." Overall, the paper displays commendable scientific merit, yet there are some points (See below) that require attention and correction to enhance its quality.

Warm regards,

Comments and suggestions

1. Please incorporate some of the important statistical values into the abstract that would greatly enhance its depth and significance. 2. In your title, two bacterial strains are mentioned (Salmonella Enteritidis and Escherichia coli). However, in your methodology section, two additional bacterial strains are introduced (Listeria monocytogenes and Staphylococcus aureus). As indicated in the results section, you employed these strains for dual-species biofilm assays or as competing species. Please clarify the specific purpose of these additional isolates in the 'bacterial strains and culture conditions' section to avoid potential contradictions with your title. 3. Line 79: "other competing species" can be revised to "selected competing species (Please specify). 4. Line 82: "Bacterial strains and growth conditions." Please include the selective agar media typically used, such as MacConkey agar, Mannitol salt agar, Listeria Oxford agar, etc., to provide a comprehensive overview of the growth conditions. 5. Line 92. …..Template plasmid pKD4, FLP helper plasmid pCP20Please specify the source of the plasmids. 6. Please ensure consistency in terminology usage across the manuscript, such as "OD 600 nm" (Line 124) and "polypropylene (PP)" (Line 293 – should be abbreviated in Line 283 where it is first mentioned). 7. Line 137: In Table 1, a list of primers for confirming the mutants is mentioned. What is the source of these primers? Have you designed it? If so, could you specify the software used for their design? If not, kindly specify the source of the primer. 8. Line 163: "Eliminating Antibiotic Resistance Gene." Could you clarify the purpose of this experiment and why specifically target one gene (FRT-flanked resistance gene)? This section requires further elaboration, and it would be beneficial to integrate the results of this experiment into the overall results. 9. Please check the position of Figure 1. If it is your figure, it should be displayed under the results section. 10. Have you stated the mechanism of how you performed Epifluorescent images in the methodology section (Parameters)? Please check? 11. The results from the "Biomass quantification by the CV Assay" are not explicitly outlined in the results section. Please review and ensure their inclusion for clarity. 12. Line 231: "Determination of fitness and Malthusian parameter." The reference is not provided. 13. Line 327. The top images show the of S. Enteritidis and the bottom images show the E. coli. Please revise it. (The top images show S. Enteritidis, and the bottom images show E. coli) 14. Line 266. How did you measure the significance level.  15. Line 276. The mutant genotype was confirmed by PCR, (Figure 1). The PCR result is not clearly displayed rather figure 1 is photo of agar plate (phenotype).  16. Line 276: "Motility agar (Figure 1)." It would be beneficial to specify the exact type of motility agar used. Figure 1 displays a solid agar plate suggesting swarming growth. Could you please clarify the specific agar medium used for clarity? Additionally, it's commonly observed that semi-solid medium is utilized for testing bacterial strain motility. Was a semi-solid medium employed in this study? 17. Line 285. “Regarding their ability to adhere to different surfaces, no significant differences found between the ∆fliC and ∆motA mutants for both speciesWould you please specify the value and mention the significance level. 18. Line 295. “for S. Enteritidis no significant differences were found” …. Mention the level of significance. 19. Line 297a 1 to 3 log reduction being observed for the first time points (2, 4 and 6 hours) please support this finding with significance level. 20. Line 329. All images have the same scale. Specify it. 21. Line 411. Contrary to what was reported in the literature (Wang et al., 2020), in our study. Please be consistent in writing the Wang et al. (2020) [28]. 22. Line 393………………Teflon, or glass (Carvalho et al., 2023). This reference is not properly cited. Please check.

Author Response

We would like to express our appreciation to the reviewer for their careful reading of the text and for all the in-depth constructive comments and suggestions about the first version of our manuscript. We have put an earnest effort to respond to the concerns of each reviewer in detail and we are now submitting a revised manuscript.

We are confident that the changes performed are in accordance with the reviewer's requests and hope the reviewer feels that our manuscript is now acceptable for publication.

A point-by-point description of our answers to the reviewer’s comments and suggestions follows. The revised manuscript has all changes highlighted in yellow to allow better follow-up by the reviewer.

Reviewer#1

Comments and Suggestions for Authors

I hope this message finds you well. I am writing regarding the recent review of the manuscript titled “The role of flagellum and flagellum-based motility on Salmonella Enteritidis and Escherichia coli biofilm formation." Overall, the paper displays commendable scientific merit, yet there are some points (See below) that require attention and correction to enhance its quality.

Warm regards,

 Authors answer: We appreciate the reviewer's feedback, and we acknowledge the reviewer has carefully analyzed the manuscript.

Comments and suggestions

  1. “Please incorporate some of the important statistical values into the abstract that would greatly enhance its depth and significance.”

Authors answer: We appreciate the reviewer's feedback, and we acknowledge the reviewer has carefully analyzed the manuscript.

  1. “In your title, two bacterial strains are mentioned (Salmonella Enteritidis and Escherichia coli). However, in your methodology section, two additional bacterial strains are introduced (Listeria monocytogenes and Staphylococcus aureus). As indicated in the results section, you employed these strains for dual-species biofilm assays or as competing species. Please clarify the specific purpose of these additional isolates in the 'bacterial strains and culture conditions' section to avoid potential contradictions with your title.”

Authors answer: We appreciate the reviewer’s feedback, and we clarified the purpose of these additional isolates in the “bacterial strains and culture conditions” section.

  1. “Line 79: "other competing species" can be revised to "selected competing species (Please specify).”

Authors answer: We appreciate the reviewer’s correction, and, as such, we have changed the manuscript accordingly.

  1. “Line 82: "Bacterial strains and growth conditions." Please include the selective agar media typically used, such as MacConkey agar, Mannitol salt agar, Listeria Oxford agar, etc., to provide a comprehensive overview of the growth conditions”.

Authors answer: For section 2.1, regarding the growth conditions, the medium used was Tryptic Soy Agar (TSA). The selective media were only used in biofilm experiments to differentiate the bacterial species within the biofilms. In fact, we have this information in the section 2.7:

“Then CFUs were counted. For better discrimination, three selective agar media were used: MacConkey agar (Liofilchem) for distinguished E. coli and S. Enteritidis, Mannitol salt agar (Liofilchem) for S. aureus and Listeria oxford agar (Liofilchem) for L. monocytogenes.”

  1. Line 92. ……..”Template plasmid pKD4, FLP helper plasmid pCP20”. Please specify the source of the plasmids. 

Authors answer: We appreciate the reviewer’s correction, and as such, we have added the source of the plasmids to the revised version of the manuscript.

  1. “Please ensure consistency in terminology usage across the manuscript, such as "OD 600 nm" (Line 124) and "polypropylene (PP)" (Line 293 – should be abbreviated in Line 283 where it is first mentioned).”

Authors answer: The OD is first mentioned at the beginning of section 2.3 (line 119), and, as such thereafter we have used the abbreviation, as well pointed out by the reviewer. Regarding polypropylene, the reviewer has a valid point, we apologize for the mistake, and, we have then changed the manuscript accordingly.

  1. “Line 137: In Table 1, a list of primers for confirming the mutants is mentioned. What is the source of these primers? Have you designed it? If so, could you specify the software used for their design? If not, kindly specify the source of the primer.”

Authors answer: Regarding the list of primers, the reviewer has a valid point, we apologize for this missing information. We have then added this information to the manuscript.

  1. “Line 163: "Eliminating Antibiotic Resistance Gene." Could you clarify the purpose of this experiment and why specifically target one gene (FRT-flanked resistance gene)? This section requires further elaboration, and it would be beneficial to integrate the results of this experiment into the overall results”

Authors answer: This step is essential to eradicate antibiotic resistance while maintaining the genetic disruptions, validated by PCR. We took advantage of FLP recombinase encoded by pCP20 plasmid to eliminate the antibiotic resistance cassette. More precisely, the mutant strains were transformed with plasmid pCP20 and subsequent incubation at 42°C as reported (Datsenko and Wanner, 2000). The pCP20 plasmid encodes the Flp recombinase that catalyzes the recombination between the FRT sites flanking the antibiotic resistance cassettes. This way we can ensure that resistance is eliminated from the strains and will have no impact on the results. This assures a “clean” knockout. To clarify, more information was added to this section in the revised version of the manuscript.

  1. “Please check the position of Figure 1. If it is your figure, it should be displayed under the results section.” 

Authors answer: We apologize for this mistake. In fact, Figure 1 in the PDF version is in the methods section, however, in the Word version, this Figure is in the correct position in the results section.

  1. “Have you stated the mechanism of how you performed Epifluorescent images in the methodology section (Parameters)? Please check?” 

Authors answer: The epifluorescent images were obtained by PNA-FISH methodology as described in section 2.10: “PNA FISH hybridization and DAPI staining”.

  1. “The results from the "Biomass quantification by the CV Assay" are not explicitly outlined in the results section. Please review and ensure their inclusion for clarity.” 

Authors answer: The results of the ‘Biomass quantification by the CV assay’ are presented in Figures 3 C and D, and described in the text between lines 312-313. We have improved this section as suggested by the reviewer.

  1. “Line 231: "Determination of fitness and Malthusian parameter." The reference is not provided.”

Authors answer: The reviewer has reason. As such, we have added a reference regarding the determination of fitness and the Malthusian parameter.

  1. Line 327. The top images show the of S. Enteritidis and the bottom images show the E. coli. Please revise it. (The top images show S. Enteritidis, and the bottom images show E. coli)

Authors answer: In order to make clear Figure 4, we have revised the figure caption. In fact, the first line corresponds to S. Enteritidis images sensitive to the Alexa Fluor 594 molecule attached to the SalPNA1873 probe, and the second line corresponds to E. coli sensitive to the Alexa Fluor 488 molecule attached to the Ec1505 LNA/2 OMe 16 probe, as reported in the section 2.10 in the manuscript.

  1. “Line 266. How did you measure the significance level.”

Authors answer: We appreciate the reviewer’s correction, as such, we have revised this section in the manuscript.

  1. “Line 276. The mutant genotype was confirmed by PCR, (Figure 1). The PCR result is not clearly displayed rather figure 1 is photo of agar plate (phenotype).” 

Authors answer: The mutants were confirmed for both genotype and phenotype, however in the manuscript, Figure 1 only describes the results of mutant’s phenotype. To clarify, we have changed the manuscript, adding “data not shown” for PCR results (genotype).  

  1. “Line 276: "Motility agar (Figure 1)." It would be beneficial to specify the exact type of motility agar used. Figure 1 displays a solid agar plate suggesting swarming growth. Could you please clarify the specific agar medium used for clarity? Additionally, it's commonly observed that semi-solid medium is utilized for testing bacterial strain motility. Was a semi-solid medium employed in this study?”

Authors answer: The medium used for the motility test was “Nutrient Broth (NB, Liofilchem), 0.5% (w/v) glucose (Liofilchem), 0.6% (w/v) bacteriological agar (Liofilchem)” as described in section 2.5. In fact, we used a semi-solid medium in this study, since we used only 0.6% (w/v) of bacteriological agar in the medium.

  1. “Line 285. “Regarding their ability to adhere to different surfaces, no significantdifferences found between the ∆fliC and ∆motA mutants for both species. Would you please specify the value and mention the significance level.”

Authors answer: We appreciate the reviewer’s feedback, and we have added information about the significance level in the revised version of the manuscript.

  1. “Line 295. “for S. Enteritidis no significantdifferences were found” …. Mention the level of significance.” 

Authors answer: We appreciate the reviewer’s feedback, and we have added information about the levels of significance in the revised version of the manuscript.

  1. “Line 297. a 1 to 3 log reduction being observed for the first time points (2, 4 and 6 hours) please support this finding with significance level.” 

Authors answer: We appreciate the reviewer’s feedback, and we have added information about the levels of significance in the revised version of the manuscript.

  1. “Line 329. All images have the same scale. Specify it.” 

Authors answer: As shown in the legend of Figure 4, all images have the same scale (5µm). In addition, we also added the magnification of the images (1000x).

  1. “Line 411. Contrary to what was reported in the literature (Wang et al., 2020), in our study. Please be consistent in writing the “Wang et al. (2020) [28]”.

Authors answer: We appreciate the reviewer’s correction, and we have changed the reference formatting in the manuscript.

  1. “Line 393………………Teflon, or glass (Carvalho et al., 2023). This reference is not properly cited. Please check.”

Authors answer: We appreciate the reviewer’s correction, and we have changed the reference formatting in the manuscript.

Reviewer 2 Report

Comments and Suggestions for Authors

This study assessed the role of the flagellum and flagellum-based motility of S. Enteritidis and E. coli in adhesion and biofilm formation by disrupting the fliC and motA genes. The results showed that motility is not essential for biofilm formation and does not have a strong impact on bacteria fitness in the presence of a competing species.

The major concerns: The ΔfliC or ΔmotA mutants were constructed and confirmed by PCR, both mutants were non-motility phenotypically. However, the presence of flagellum of the mutants (especially the ΔmotA mutant) were not determined. So, some conclusions in this study, such as “motility, and not flagella, might be the key factor (affect the adhesion and biofilm formation)”, were not too solid.

The unintentional disruption of genes was not addressed when constructing the ΔfliC and ΔmotA mutants. Or a complementary strain was not used in this study.

Author Response

Reviewer#2

“This study assessed the role of the flagellum and flagellum-based motility of S. Enteritidis and E. coli in adhesion and biofilm formation by disrupting the fliC and motA genes. The results showed that motility is not essential for biofilm formation and does not have a strong impact on bacteria fitness in the presence of a competing species. The major concerns: The ΔfliC or ΔmotA mutants were constructed and confirmed by PCR, both mutants were non-motility phenotypically. However, the presence of flagellum of the mutants (especially the ΔmotA mutant) were not determined. So, some conclusions in this study, such as “motility, and not flagella, might be the key factor (affect the adhesion and biofilm formation)”, were not too solid. The unintentional disruption of genes was not addressed when constructing the ΔfliC and ΔmotA mutants. Or a complementary strain was not used in this study.”

Authors answer: We appreciate the reviewer's feedback and we acknowledge that the reviewer has carefully analyzed the manuscript. It is important to highlight that Figure 1 only shows the results of motility phenotypically. However, the mutants were confirmed by PCR/sanger sequencing for both motility and flagella genetic regions. Indeed, since no significant differences were found between the ∆fliC and ∆motA mutants for both species regarding their ability to adhere to different surfaces, we conclude that it is motility, and not flagella, might be the key factor, otherwise difference would be noticed between the two mutants for their initial adhesion and biofilm formation (Figure 2).

Reviewer 3 Report

Comments and Suggestions for Authors

The ability of microorganisms to form a biofilm is an important issue for many fields. Research on this issue is important and constantly relevant. Below are some comments to the authors:

Material:

- please use abbreviations, i.e. h, min, etc

- line 256 - is this fig 1 ? please add description and data to the text. whether the description of this fig is in the lines 279-280 ?

Author Response

Reviewer#3

“The ability of microorganisms to form a biofilm is an important issue for many fields. Research on this issue is important and constantly relevant. Below are some comments to the authors:

Material:

“- please use abbreviations, i.e. h, min, etc”

Authors answer: We appreciate the reviewer's feedback, and we have changed the revised version of the manuscript accordingly.

“- line 256 - is this fig 1 ? please add description and data to the text. whether the description of this fig is in the lines 279-280 ?”

Authors answer: We appreciate the reviewer's feedback, and, as such, we have added additional information regarding Figure 1 to the revised version of the manuscript.

Round 2

Reviewer 2 Report

Comments and Suggestions for Authors

As response, Figure 1 showed the results of motility phenotypically and the mutants were confirmed by PCR/sanger sequencing for both motility and flagella genetic regions, and no significant differences were found between the ∆fliC and ∆motA mutants for both species regarding their ability to adhere to different surfaces, so authors conclude that it is motility, and not flagella, might be the key factor, otherwise difference would be noticed between the two mutants for their initial adhesion and biofilm formation.

If the ∆fliC strains showed the absence of flagella, while the âˆ†motA mutants showed flagella there, the conclusion would be solid.

Author Response

We would like to express our appreciation to the reviewer for their careful reading the revised version of the manuscript, and for the comment.

We are confident that the changes performed are in accordance with the reviewer's requests and hope the reviewer feels that our manuscript is now acceptable for publication.

The revised manuscript has all changes highlighted in green to allow better follow-up by the reviewer.

Comment:

“As response, Figure 1 showed the results of motility phenotypically and the mutants were confirmed by PCR/sanger sequencing for both motility and flagella genetic regions, and no significant differences were found between the âˆ†fliC and âˆ†motA mutants for both species regarding their ability to adhere to different surfaces, so authors conclude that it is motility, and not flagella, might be the key factor, otherwise difference would be noticed between the two mutants for their initial adhesion and biofilm formation.

 If the âˆ†fliC strains showed the absence of flagella, while the âˆ†motA mutants showed flagella there, the conclusion would be solid”

Author’s answer: The reviewer has a valid point here. In fact, we did not show that the ∆fliC strains showed the absence of flagella, while the âˆ†motA mutants showed flagella. Thus, we have changed the manuscript (lines 464-465) in order to show our hypothesis, based on our results, instead of a solid conclusion since we are aware that we have this limitation in our study.